# Design of a Tension Infiltrometer with Automated Data Collection Using a Supervisory Control and Data Acquisition System

**DOI:** 10.3390/s23239489

**Published:** 2023-11-29

**Authors:** David Alberto Morales-Ortega, Víctor Hugo Cambrón-Sandoval, Israel Ruiz-González, Hugo Luna-Soria, Juan Alfredo Hernández-Guerrero, Genaro García-Guzmán

**Affiliations:** Natural Sciences and Engineering Faculty, Airport and Cerro de las Campanas Campus, Universidad Autónoma de Querétaro, Santiago de Querétaro 76010, Mexico; dmorales69@alumnos.uaq.mx (D.A.M.-O.); israel.ruiz@uaq.edu.mx (I.R.-G.); hugoluna@uaq.mx (H.L.-S.); juan.hernandez@uaq.mx (J.A.H.-G.); genaro.garcia@uaq.mx (G.G.-G.)

**Keywords:** infiltration, SCADA system, automated instrumentation

## Abstract

This study highlights the importance of water infiltration in hydrological basin management, emphasizing its role in water services, water quality regulation, and temporal patterns. To measure this crucial function, this study introduces a portable and user-friendly tension infiltrometer designed for easy assembly and data collection. The tension infiltrometer, based on the 2009 design by Spongrová and Kechavarzi, offers a comprehensive characterization of the soil properties related to water flow. It eliminates the influence of preferential flow, providing accurate data. Additionally, it accommodates changes in pore size distribution within the soil, which is crucial for understanding water movement. This study discusses the challenges associated with traditional infiltration measurement tools, like double-ring infiltrometers and single rings, which are not easily transported and can lead to inaccuracies. In response, the proposed infiltrometer simplifies data collection, making it accessible to a broader range of users. This study also explores the use of the VL53L0X distance sensor in the infiltrometer, providing an innovative solution for measuring the water column height. The system’s user interface allows real-time data collection and analysis, significantly reducing the processing time compared to that of the manual methods. Overall, this work demonstrates the potential for advancement in hydrological basin management using user-friendly instrumentation and automated data collection, paving the way for improved research and decision making in environmental services, conservation, and restoration efforts within these ecosystems.

## 1. Introduction

Water infiltration plays a crucial role in managing hydrological basins [1]. It serves as a key ecosystem function that is recognized for its contributions to water services, water quality regulation, and temporal patterns. This function is intricately linked to the state of vegetation cover and soil conditions [2]. Globally, it has been established as an indicator of environmental services, making it essential for managing the payments related to watershed conservation and restoration, as well as for making future projections [3].

Some researchers use saturated hydraulic conductivity (Ks) as a descriptive variable for this function. Ks allows comparisons among sites with varying initial moisture levels and different characteristics of vegetation and soil cover [4]. However, measuring Ks in situ can be challenging because the existing estimation methods rely on variables that can be difficult to establish or estimate. This challenge is exacerbated by the use of analogue tools, like double-ring infiltrometers or single rings, which were originally designed for different purposes. These tools are not easily transported to regions with a steep topography, resulting in longer data processing times, increased transportation costs, and potential inaccuracies due to human factors [5].

The need for a portable and user-friendly instrumentation model to monitor infiltration data across various basins arises from the necessity to adhere to standardized procedures for environmental services payments and to guarantee precise data collection during field evaluations.

To address this need, the INDI-INECOL infiltrometer [6], based on the 2009 design by Spongrová and Kechavarzi, was chosen as a reference instrument. Tension infiltrometers provide a comprehensive characterization of various hydro physical properties essential for understanding water flows in soil. They accurately measure the flow of water entering the soil, eliminating the influence of preferential flow that often occurs under saturated conditions. These instruments also enable the characterization of the water conduction capacity in pores of different sizes, including active macro- and mesopores, by establishing a connection between the soil’s pore space and a water reservoir.

Soil functions as a porous medium, with pores that are partially filled with water and air. When it comes to water movement, these pores can be represented by capillary tubes with circular sections defined by their radius. The height at which water rises inside these tubes due to capillary forces is inversely proportional to the tube’s radius. Similarly, in soil, the capillary suction value of smaller pores is greater than that of the larger ones. If the capillary suction (S) value of the soil is equal to or less than the tension (T) exerted with the infiltrometer on its surface (S ≤ T), no flow occurs from the infiltrometer reservoir to the soil, resulting in a flow volume per unit of time or an expense (Q) of zero. Changes in land use can significantly impact porosity, leading to substantial alterations in the infiltration process [7].

To simplify the interpretation and analysis of infiltration data, a cost-effective instrument with easy assembly is proposed. While prior work in this field often required specialized expertise to understand procedures and results, this proposed instrument can be utilized by individuals interested in evaluating and managing environmental services related to the infiltration process in hydrological basins with varying land uses.

## 2. Materials

### 2.1. Mechanical Pieces

Three new pieces were used to replace the base and Mariotte tube as well as main reservoir support mentioned in the *Infiltrometry Manual* [6]. The design was intended for an easy assembly system, modifying the connectors into a pressure assembly and replacing the metal ball valve with a PVC one. They were 3D printed and made of a hard resin polymer.

The third piece was designed for supporting the control system, and it was made of PET Polymer.

#### 2.1.1. Infiltrometer Base

The base was designed with mechanical elements from the base of the INDI-INECOL infiltrometer. This included a water distribution chamber, which was designed with a smaller volume to reduce the drop in the water column when opening the valve and starting rehearsal; a 36 mm diameter connection to press-fit the 1-inch PVC ball valve and another 11 mm diameter connection to connect a rubber tube to the Mariotte system.

The diameter of contact with the ground was modified to improve stability in the field, without modifying the diameter of the water outlet, adding to the design. An edge under the base was included to reduce the loss of contact with the ground during the operation of the equipment. Finally, the perforated bottom with 2 mm holes (25) was attached to the base design, and 4 fastening points were added to the outside of the piece, which allowed the infiltrometer to be staked when performing the tension test. Figure 1 shows the scheme and views of the described component.

#### 2.1.2. Connection for Reservoir and Mariotte

The Mariotte holder was designed to be able to be pressed directly into the PVC valve, replacing the metal ball valve from the original design and supporting the water column weight at the reservoir and Mariotte levels. The design included connections for the Mariotte conduit to the base and custom holes used to insert the acrylic tubes (Mariotte and main reservoir) to pressure-assembly them. Figure 2 shows a scheme of the components described.

#### 2.1.3. Sensor Support

A piece was designed to hold the physical components of the SCADA system, where an Arduino Uno card coupled to a commercial acrylic casing was attached to this piece called the sensor support, and it was designed to be placed in the water inlet of the main reservoir at the top of the infiltrometer. Figure 3 shows a scheme of the described component.

### 2.2. Electronic Items for SCADA System

The SCADA system was developed with the following physical components: an Arduino Uno control card coupled to a Time-of-Flight (ToF) sensor, VL53L0X. For computational control, the Arduino Ide is used to read the sensor data and generate a message on the serial port with the distance records in millimeters. Matlab R2023a version was used to read this serial port and develop a graphical user interface for displaying, storing, and enriching the sensor data. The data were finally exported as files suitable for Microsoft Excel Profesional Plus 2019 version.

### 2.3. Sensor Assembly

The infiltrometer assembly design is simple, so that it can be built in situ. For this, an assembly diagram was made for, which shows the sensor and rubber plug assembly sequence. The scheme can be found in the Appendix A.

### 2.4. Infiltrometer Installation to Surface

To install the instrument on the ground, it is necessary to follow the steps in Figure 4, which considers the nylon mesh mentioned in the bill of materials list, and a contact surface, that in our case was a fine marble sand.

## 3. Experimental Program of the Automated Data Collection System, SCADA

The SCADA system is defined as a control, action, decision, storage, and data recording system. This system collects field data from a sensor connected to a master station, which, in our case, is an Arduino Uno card, for reading through the serial port. Matlab was used to read this serial port, and a graphical user interface was developed for the display, storage, and enrichment of the sensor data. The data were finally exported as files suitable for Microsoft Excel.

A VL530LX sensor was selected that is based on a Time-of-Flight (ToF) distance measurement system, which makes it possible to accurately measure the time it takes for light to travel from the closest object and return to the sensor. It has a supply voltage of 3 V–5 V DC, a current of 10 mA (40 mA Max), a resolution of 1 mm, and works at an operating temperature from −20 °C to 70 °C. Its dimensions are 25 mm × 10.3 mm × 3.5 mm. This sensor detects the height of the water column using a circle of Styrofoam that floats in the water column within the main reservoir, which descends as the soil consumes the water from the reservoir.

### 3.1. Programming the Data Collection through Ide Arduino

To program the VL53L0X sensor, it was downloaded from the Adafruit library, and it continued to adapt to the measurement ranges by choosing to measure every 10 s (100,000 ms). The code is shown in Appendix B.

### 3.2. User Interface in Arduino and Matlab Environment

Matlab [8] was used to read the data through the serial port and through a graphical user interface developed using Appdesigner within the same environment for the display, storage, and enrichment of the sensor data to finally be exported as appropriate MS Excel files.

In this environment, we can use apps that are stand-alone programs with a GUI graphical user front end that automates a task or calculation. The programmed application collects data in real time on the distance of the water column within the main reservoir. The application was designed with four buttons, a table (Table), and an axis graph (Axes), which perform the following functions:⚬The Start button (Start). When pressed, the reading of the data from the serial port begins, and from that moment, the system memory stores the data, ordering them into a table component (Table) and storing them as virtual memory to also graph them (Axes), so that, along the ordinates, the collection time is plotted in hours, minutes, and seconds against the height of the water column in mm along the abscissas.⚬The end of data collection button (Stop). When this button is pressed, it ends the functions of reading data in the serial port, as well as graphing and ordering in the table.⚬A button exports the data obtained (Export) to a suitable file for Microsoft Excel for further processing.

The work program was carried out using the Matlab Appdesigner application, as is shown in Appendix C.

### 3.3. Calibration of the Measurement System

The calibration of the sensor was carried out through a comparison with a tape measure placed in a rail-type system, in which a Styrofoam wall (white) was placed perpendicular to the rail, with the possibility of moving up to 1000 mm on it and with a fastening for the sensor at one end. It used this to detect the wall at the distances chosen for the test, to carry out 30 measurements (one per second) at distances of 100, 200, 300, 400, 500 and 600 mm, and calculate the standard deviation, as well as the absolute and relative errors for each value.

Once the program was chosen in the Arduino environment, it was uploaded to the Uno board, and sensor calibration was performed. The increased measurement error turned out to be acceptable in the range between 0 and 400 mm, taking into consideration that, in the calculation, 6, 3.5, 2 and 11.5% were obtained; however, when this distance increased, the error increased up to 37%. Thus, it is not pertinent to carry out measurements for values greater than 400 mm. These results are shown in Figure 5.

## 4. Results

The instrument test was carried out at the Autonomous University of Querétaro facility, showing that the objectives of the mechanical infiltrometer of improving its portability, simplifying the construction, and improving the hermeticity of the instrument were achieved. Data collection, in which the user interface was designed and tested, allowed the validation of the objective of obtaining quality data in real time and reducing the data processing time.

Once the working tension was selected (size of porosity to test), the Mariotte was filled with water to the selected height in centimeters. Then, the main reservoir was filled, leaving a minimum empty distance of 3 cm with respect to the nozzle of the main reservoir. Once the above steps had been completed, the cap of the main reservoir, previously assembled with the sensor support and the SCADA system, was placed and connected to the field computer using a USB cable. It is necessary to run the system application, in this case, it is named AppIndi, which will be visually displayed on the screen, as shown in the design. After deployment, with the instrument installed in the ground, data collection must be initiated by pressing the Start button and immediately opening the valve of the infiltrometer. As the water column descends, it will be graphed in real time. However, it is recommended that the measured working height does not exceed 40 cm, or 400 mm, as the measurement error of the VL53L0X sensor exponentially increases. In such cases, data collection should be stopped by pressing Stop to refill the reservoir. Once the reservoir is refilled, the Append button must be pressed to store the previously recorded data and prevent data loss. Afterward, the Start button must be pressed again to restart data collection. To avoid data loss, it is advisable to export the data to Excel immediately after the tension test has been performed. The following figure shows the application display after the tension test at the university. It displays a blue dotted line, which represents data over time, and a red dotted line, which is the result of pressing the Calculate K button. This button calculates the true data as a straight line and displays the equation y = mx + b in the text box. Hydraulic conductivity is represented by the slope of the line in mm/hr.

Hydraulic conductivity for the tension test at a tension of −7.0 cm is given by the Equation 0.86247X + 104.5546, which means that our value of conductivity at the time of the experiment is 0.8624 mm/h. This can be observed in Figure 6.

During this time, the comparison of the data collection results was conducted using a double-ring infiltrometer and an automated tension infiltrometer. After manual data collection, it took 0.85 h to conduct the manual data transfer to Excel for each collection, involving the transfer of 180 data points to plot the descriptive curve. However, this time was reduced to 0.01 h when automatically exporting the data to Excel, instantly obtaining the descriptive equation. This allows the calculation of hydraulic conductivity, followed by the determination of both the saturated and unsaturated hydraulic conductivities.

## 5. Discussion

For developing the instrument, a series of actions were undertaken, leading to the results presented in this work. Firstly, the choice of a tension infiltrometer was made due to the necessity of understanding the relationship between vegetation cover and soil conditions, including the shallowest soil horizon. The analysis of various infiltration measurement instruments, such as double-ring infiltrometers, revealed that their operation required the modification of the soil’s organic horizon and insertion, preventing the observation of the desired relationship. In this study, tests were conducted using a portable double-ring instrument, addressing the challenges in its design.

While a replica of a tension infiltrometer was created following the *Infiltrometry Manual* mentioned earlier, it allowed the observation of mechanically improvable characteristics based on the principle of replicability. The parts needed to be simplified for easy assembly by anyone. It was determined that the parts should be 3D printed using polymers, enabling those interested in replication to obtain the electronic file and print it with any waterproof material printing service.

Following the resolution of mechanical design details, the development of automated instrumentation commenced. It was based on previous automations, where an analogue pressure difference sensor (MXP5010DP model) was employed to measure the water column. However, for the proposed infiltrometer, this sensor did not achieve its objective during the system tests, as it could not correlate the negative pressure within the main reservoir. Nevertheless, this possibility was not ruled out.

A decision was made to change the sensor and measure the water column height using a distance sensor. An ultrasonic sensor was initially used; however, due to the size of the main reservoir and the sensor’s operating principle, it was difficult, and the water column height could not be correlated with the sensor.

The VL53L0X (ToF) distance sensor, as presented in the results, was selected to meet the size and functionality requirements. It demonstrated excellent correlation results between the measured distance and the water column height, leading to its adoption.

It is important to note that the SCADA system was developed with accessible elements to facilitate assembly through part acquisition. Therefore, with the correct software, it is possible to replicate it. For data collection using the VL53L0X sensor, it is recommended that users have a working column of 450 mm to maintain a maximum acceptable error of 12% in the measurement. Additionally, considering the energy consumption of the PC used for data collection, having an external battery supply is highly recommended for longer tests.

The work by Gómez Tagle et al. [9] which involves the automation of a tension infiltrometer, enables data collection and linear adjustment calculations to determine Ks. However, the components used in their work are less accessible to a general user, as the datalogger used for control is more robust and costly. The proposed SCADA system in this work is an accessible solution that remains open for improvement, whether in terms of data quality through sensor upgrades or by employing techniques using the same control system to enhance the ecohydrological evaluation, such as detecting vertical and lateral infiltration processes through electrical conductivity.

## 6. Conclusions

The increasing use of automated systems for collecting ecosystem data based on development boards offers significant opportunities for science in terms of research and data acquisition for decision making. From a watershed perspective, it provides a broader view of the requirements for sustaining ecosystem functions and services, such as infiltration.

As a starting point, proposing the instrumentation design for research makes it possible to envision the development of technology that can provide more precise data on water in its various pathways within the hydrological cycle. These data can be integrated to enhance the information gathered using a single instrument. This would enable, from the perspective of integrated watershed management, the creation of a path for acquiring the necessary knowledge for the management of natural resources, restoration, and social projects within these territories.

## Figures and Tables

**Figure 1 sensors-23-09489-f001:**
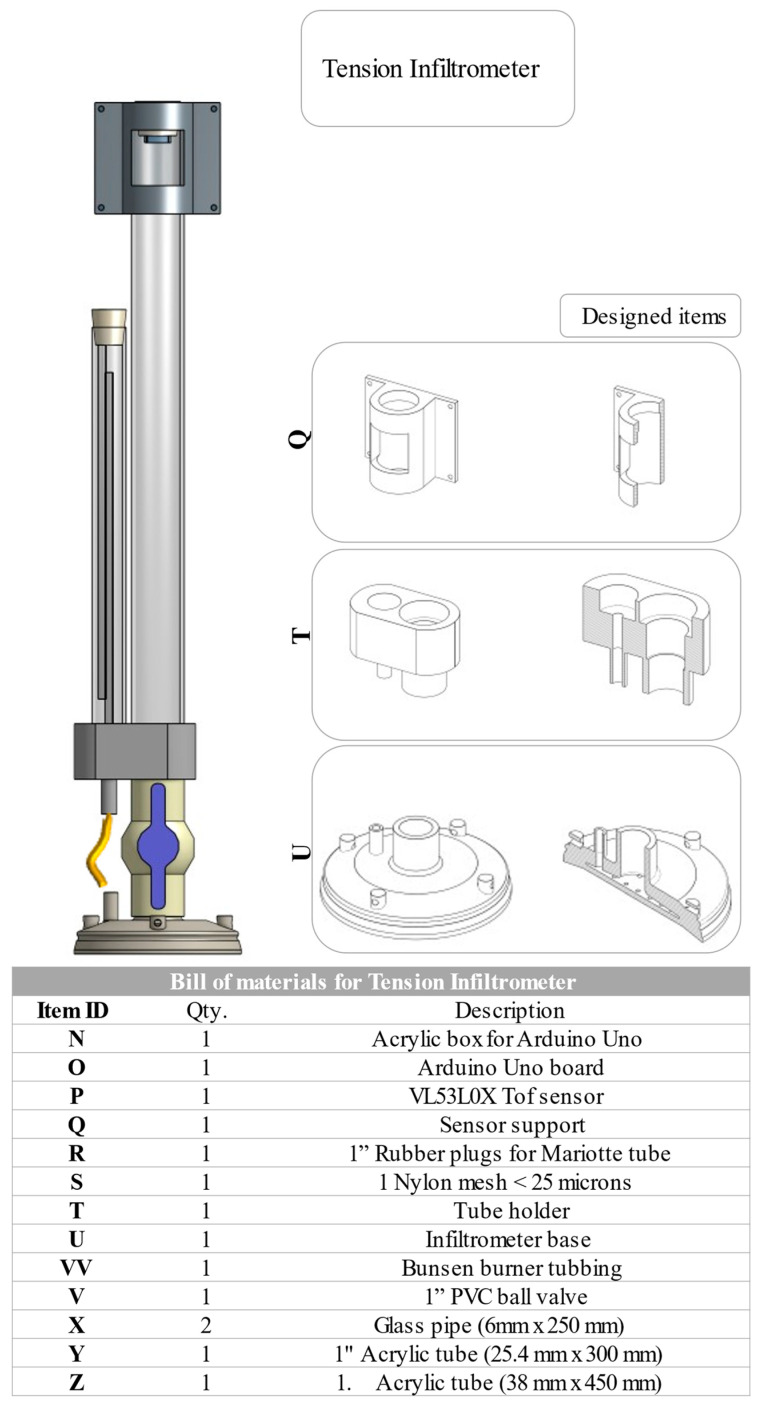
Elements of the tension infiltrometer, list of materials, and designed components.

**Figure 2 sensors-23-09489-f002:**
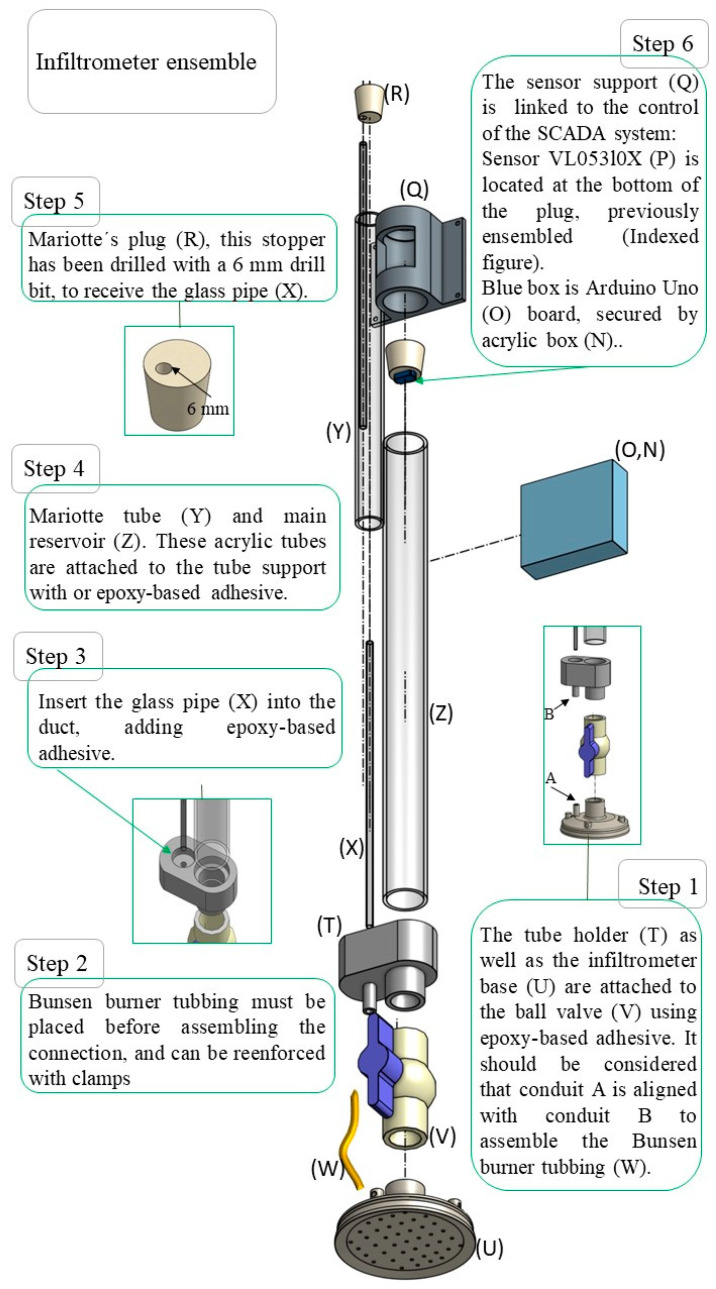
Assembling the tension infiltrometer.

**Figure 3 sensors-23-09489-f003:**
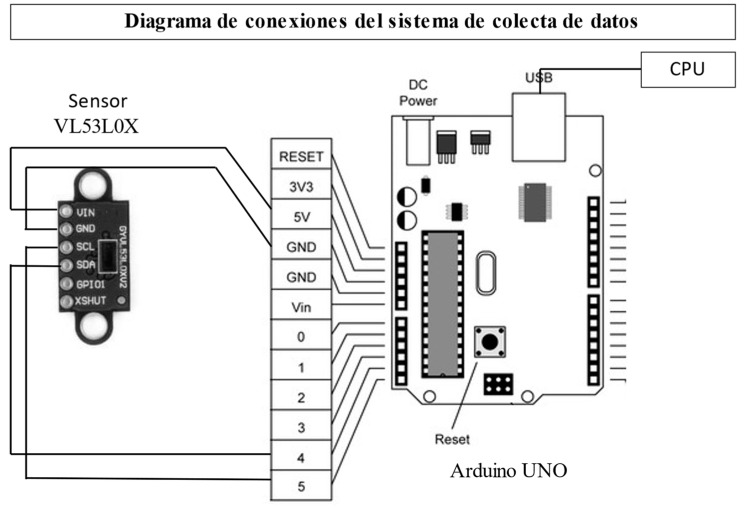
Scheme of infiltrometer automated data collection system and its connections.

**Figure 4 sensors-23-09489-f004:**
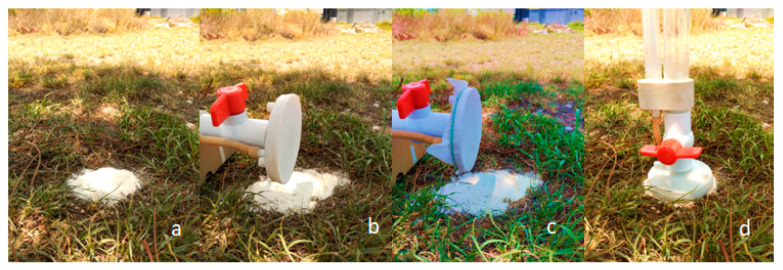
Installation procedure, where (**a**) the contact surface is located on the ground. (**b**) Infiltrometer base without nylon mesh. (**c**) Infiltrometer base with nylon mesh. (**d**) Infiltrometer installed on the ground.

**Figure 5 sensors-23-09489-f005:**
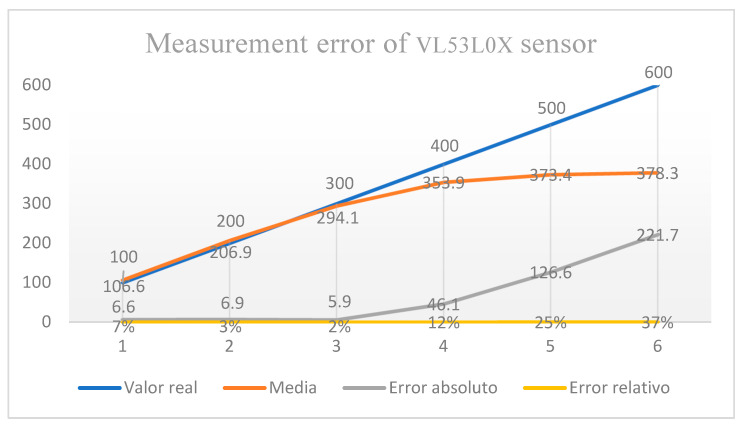
VL053L0X results.

**Figure 6 sensors-23-09489-f006:**
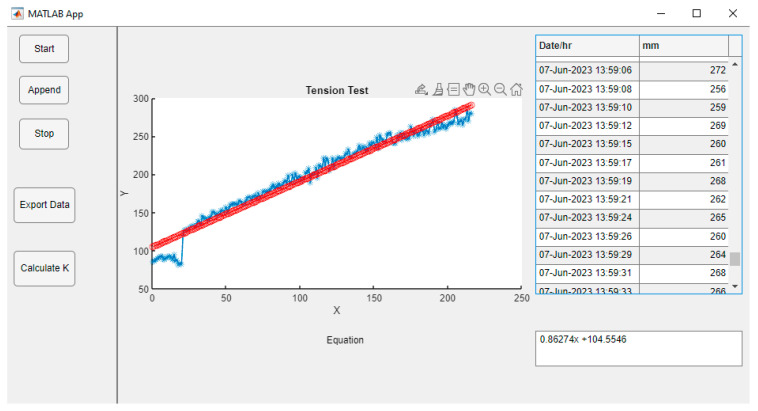
Tension test results. Screenshot of appINDI display. The blue line shows the data collected on site and the red line the linear regression that the app performs when calculating the hydraulic conductivity.

## Data Availability

The results can be observed in the Diagnosis of infiltration under different land uses and vegetation in La Beata micro-basin, Querétaro, 2023 Magister Thesis, Universidad Autónoma de Querétaro, Natural Sciences Faculty.

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
