# Peer review of "Design of a Tension Infiltrometer with Automated Data Collection Using a Supervisory Control and Data Acquisition System"

_sensors, 2023, doi:10.3390/s23239489_

Round 1

Reviewer 1 Report

Comments and Suggestions for Authors

Dear Authors,

Although the topic presented is important, unfortunately I could not find the presentation done properly. The whole manuscript must be improved with respect to all sections for further considerations. Line number 157-159, what is this about? There are many such problems in the manuscript, which drastically reduces the readability. 

Best regards

Comments on the Quality of English Language

Long and complex sentences that are difficult to comprehend must be simplified

Author Response

Dear reviewer,

We appreciate and appreciate your comments on our manuscript, so what we did was review its structure, and made the appropriate corrections for the grammar, fluency, and clarity of the text.

The purpose of the project has been clarified in the introduction, to highlight the relevance of this type of instruments for the management of hydrological basins.

Best regards. 

Reviewer 2 Report

Comments and Suggestions for Authors

Dear colleagues! This is a good and up-to-date development that helps automate the collection of data on the hydraulic properties of soils. There are a number of comments in the text that, from our point of view, will help improve the article. The main problems are a small section "Discussion", it is desirable to expand it, as well as confusion with units of measurement. The infiltrometer measures the conductivity in mm/time. And we need to use these units, not just mm. Or explain that the measurement error of the sensor depends on the height of the water in the infiltrometer (m), and not on the estimated intensity of its movement (mm / time) (Figure 5). Please make a minor revision of the manuscript, solving these problems.

August 24. 2023.

Comments on the Quality of English Language

The language is quite understandable, but there are typos and the use of Spanish instead of English.

Author Response

Dear reviewer

We appreciate and appreciate your comments on our manuscript, so what we did was review its structure, for a better undestanding of the structure of the manuscript and made some improvements for the grammar, fluency, and clarity of the text.

Discusion and conclusion sections has been improved to give more detail about our experience at this project.

Best regards.

Reviewer 3 Report

Comments and Suggestions for Authors

The authors designed a tension infiltrometer with automated data collection with SCADA. The manuscript is interesting and well-written. However, there are a few minor comments which help to improve the manuscript.

1.      The entire abstract section looks vague and requires comprehensive overhaul. The specific objectives of the study and the pertinent qualitative and quantitative research findings have to be briefly addressed.

2.      Lines: 51-57. How do the testing conditions affect the determination of infiltration conditions and how does it affect the overall performance? The authors are expected to refer to the following paper and address this issue while revising the manuscript. (https://doi.org/10.3390/ma14113120)   

3.      The end paragraph of the study requires does not give any idea about the proposed objectives of the work. Lines: 95-99 requires comprehensive redrafting by bringing out the gaps identified in earlier works and the scope of the research undertaken. Consider revising it comprehensively.

4.      The materials and methods section requires comprehensive overhaul. Consider redrafting it completely.

5.      Fig 1 should appear in Section 2. Revise it accordingly.

6.      The present study is based on a specific design of a tension infiltrometer and a SCADA system, the findings may not be easily applicable to other setups or systems. This limits the generalizability of the study's results. Clarify this. Provide a valid rationale critique by relying on existing literature how the test results from the current study can be envisaged to be used to new novel works.

7.      Lines 214-242 can be provided in Appendix and cited accordingly in the text. Revise it.

8.      Remove Fig 4. The GUI interface is not relevant to be provided.

9.      Provide work program (Lines 271-496) is the appendix and rewrite these sections accordingly. It cannot be a part of the main article.

10.  Provide some takeaways using this work program for different testing conditions as an output in the Appendix section so that, the other researchers might actually understand the code.

11.  How to prevent data loss in case of system failures or technical issues? How does it affect the overall result and what counter measures the authors propose to be incorporated in such an eventuality?

12.  Provide a comparative analysis from existing other systems to SCADA system and highlight the importance of present work.

13.  How the design and implementation of the infiltrometer and SCADA system are practical for field applications? Were there any practical limitations or challenges encountered during the study?

14.  The discussion section looks pretty weak and requires comprehensive rewriting. Consider revising the entire section.

15.  The conclusions are not at all technically strong. Consider revising them all by highlighting the qualitative and quantitative takeaways from the work carried out.

Author Response

Dear reviewer,

We appreciate and appreciate your comments on our manuscript, so what we did was review its structure, and made the appropriate corrections for the grammar, fluency, and clarity of the text.

The purpose of the project has been clarified in the introduction, to highlight the relevance of this type of instruments for the management of hydrological basins.

The abstract section has been improoved.

The testing conditions what we are boarding it sil´s porosity, so we describe it at introduction.

We have made an overhaul to materials, experimental program and results section to make sense of wwhat we have done.

All the programming lines, has been moved to Appendix A and B.

The discusion section has been improved speaking about the pertinence and importance of this work, and the conditions that had solve while doing it.

We hope our conclusion can be undestood as the importance that technology could had by helping improving watersheads management.

Best regards!

Round 2

Reviewer 1 Report

Comments and Suggestions for Authors

Please improve "English"

Comments on the Quality of English Language

Please improve "English"

Reviewer 3 Report

Comments and Suggestions for Authors

The authors must provide one by one response sheet to each of the queries raised. It seems, the authors have relied on giving pretty simple and vague answers which are not at all acceptable when taking up comprehensive review. Accordingly, the authors must revisit the comments raised during first review by this reviewer and provide line by line response sheet and show by means of line numbers where such corrections are incorporated in the revised manuscript. 

1.      The entire abstract section looks vague and requires comprehensive overhaul. The specific objectives of the study and the pertinent qualitative and quantitative research findings have to be briefly addressed.

2.      Lines: 51-57. How do the testing conditions affect the determination of infiltration conditions and how does it affect the overall performance? The authors are expected to refer to the following paper and address this issue while revising the manuscript. (https://doi.org/10.3390/ma14113120)   

3.      The end paragraph of the study requires does not give any idea about the proposed objectives of the work. Lines: 95-99 requires comprehensive redrafting by bringing out the gaps identified in earlier works and the scope of the research undertaken. Consider revising it comprehensively.

4.      The materials and methods section requires comprehensive overhaul. Consider redrafting it completely.

5.      Fig 1 should appear in Section 2. Revise it accordingly.

6.      The present study is based on a specific design of a tension infiltrometer and a SCADA system, the findings may not be easily applicable to other setups or systems. This limits the generalizability of the study's results. Clarify this. Provide a valid rationale critique by relying on existing literature how the test results from the current study can be envisaged to be used to new novel works.

7.      Lines 214-242 can be provided in Appendix and cited accordingly in the text. Revise it.

8.      Remove Fig 4. The GUI interface is not relevant to be provided.

9.      Provide work program (Lines 271-496) is the appendix and rewrite these sections accordingly. It cannot be a part of the main article.

10.  Provide some takeaways using this work program for different testing conditions as an output in the Appendix section so that, the other researchers might actually understand the code.

11.  How to prevent data loss in case of system failures or technical issues? How does it affect the overall result and what counter measures the authors propose to be incorporated in such an eventuality?

12.  Provide a comparative analysis from existing other systems to SCADA system and highlight the importance of present work.

13.  How the design and implementation of the infiltrometer and SCADA system are practical for field applications? Were there any practical limitations or challenges encountered during the study?

14.  The discussion section looks pretty weak and requires comprehensive rewriting. Consider revising the entire section.

       15.  The conclusions are not at all technically strong. Consider revising them all by                   highlighting the qualitative and quantitative takeaways from the work carried out.

Only then, it can be considered a serious address of review comments.

Round 3

Reviewer 3 Report

Comments and Suggestions for Authors

The authors have addressed my comments raised during the second review. The reviewer is satisfied with their response and recommends acceptance of the revised version of the manuscript.